# Effectiveness of a Remote Monitoring-Based Home Training System for Preventing Frailty in Older Adults in Japan: A Preliminary Randomized Controlled Trial

**DOI:** 10.3390/geriatrics9010020

**Published:** 2024-02-18

**Authors:** Yasuhiro Suzuki, Yukiyo Shimizu, Yuichiro Soma, Takaaki Matsuda, Yasushi Hada, Masao Koda

**Affiliations:** 1Department of Rehabilitation Medicine, University of Tsukuba Hospital, Tsukuba 305-8576, Ibaraki, Japan; 2Tsukuba Therapist Society for Diabetes Mellitus Prevention, Tsukuba 305-8576, Ibaraki, Japan; 3Biomedical Science and Engineering Research Center, Hakodate Medical Association Nursing and Rehabilitation Academy, Hakodate 040-0081, Hokkaido, Japan; 4Department of Internal Medicine (Endocrinology and Metabolism), Institute of Medicine, University of Tsukuba, Tsukuba 305-8576, Ibaraki, Japan; matsuda.takaaki.dd@ms.hosp.tsukuba.ac.jp; 5Department of Rehabilitation Medicine, Institute of Medicine, University of Tsukuba, Tsukuba 305-8575, Ibaraki, Japan; shimiyukig@md.tsukuba.ac.jp (Y.S.); y-hada@md.tsukuba.ac.jp (Y.H.); 6Department of Orthopedic Surgery, Institute of Medicine, University of Tsukuba, Tsukuba 305-8576, Ibaraki, Japan; ysouma@times.hosp.tsukuba.ac.jp (Y.S.); masaokod@gmail.com (M.K.)

**Keywords:** lower-load resistance training, squats, muscle hypertrophy, sarcopenia, SUKUBARA^®^, frailty

## Abstract

This study examined whether SUKUBARA^®^, a remotely managed training system that we developed, could improve skeletal muscle mass and muscle strength in community-dwelling older adults. SUKUBARA^®^ is a composite exercise program that combines lower-load resistance training and balance exercises. Participants were instructed to exercise while watching individually assigned videos on YouTube, such that the research administrators could verify the viewing records of each participant. Fifteen participants (69 ± 4 years) were randomly assigned to the intervention (eight participants; the RT group) or the control group (seven participants; the CO group). The primary endpoint was a change in fat-free mass (FFM; kg), whereas the secondary endpoints included a change in knee extension strength (KES; Nm/kg). Correlation analyses were conducted to examine the relationship between FFM and KES. During the 12-week intervention period, significant differences were observed between the RT and CO groups in the changes in FFM (0.5 ± 0.5 vs. −0.1 ± 0.5) and KES (0.20 ± 0.22 vs. 0.02 ± 0.13), and significant positive correlations were found between the changes. Thus, SUKUBARA^®^-based interventions have the potential to improve muscle hypertrophy and enhance muscle strength among community-dwelling older adults. Thus, SUKUBARA^®^ -based interventions show promise in improving muscle hypertrophy and enhance muscle strength among community-dwelling older adults. However, appropriately powered future research is needed to replicate these findings.

## 1. Introduction

Japan is facing a severe health crisis due to its aging population, with frailty being the pertinent condition. Frailty critically challenges the extension of healthy life expectancy, since it is associated with a high risk of long-term care dependency and mortality [1,2,3]; however, an improvement in functionality is possible. Thus, addressing frailty is a major public health challenge in the 21st century. Additionally, sarcopenia, which refers to the age-related decline in skeletal muscle mass (SMM) and function, is recognized as a central component of physical frailty [4]. In 2019, the Asian Working Group for Sarcopenia (AWGS) published revised criteria for sarcopenia in Asia, including low SMM in combination with either low muscle strength or low physical function [5]. Recent meta-analyses have reported that sarcopenia is associated with negative health outcomes such as reduced physical performance and increased mortality [6,7]. Therefore, it is imperative to prevent frailty and sarcopenia to maintain independence and quality of life in older persons without increasing their economic burden. 

The most effective prescription for increasing SMM and strength in older adults is resistance training [8,9,10]. Particularly, resistance exercises performed at around 70% of one’s maximum lifting capacity (one repetition maximum strength; 1RM) are effective for muscle hypertrophy [11]. However, increasing evidence suggests that lower-load resistance training (LLRT), that is, training with loads below 50% of 1RM performed to near muscle failure, may be an effective alternative to traditional high-load training and, in many cases, promote similar or even superior physiological adaptations including muscle hypertrophy [12,13,14,15]. Therefore, LLRT implementation offers specific advantages for individuals at risk, such as older adults or those with chronic conditions. 

In a large-scale study targeting community-dwelling older adults, significant muscle hypertrophy effects were reported with LLRT (around 30–50% of 1RM; 8–13 repetitions × three sets) conducted at home [16]. This method is highly compatible with a home-based exercise intervention since it requires minimal equipment and no special facilities. However, to widely promote LLRT as a self-training method for community-dwelling older adults at home, several challenges related to existing exercise protocols need to be addressed. To elaborate, simplifying the process by minimizing the variety of exercise programs is essential to establish a remote healthcare system that allows for both convenient and accurate monitoring of exercise practices. Therefore, we previously developed a home training system that incorporates an LLRT concept called SUKUBARA^®^, which combines a simplified exercise protocol with a remote monitoring system [17]. SUKUBARA^®^ utilizes a video viewing method wherein personalized YouTube videos are created for each participant. We previously conducted a pilot study with younger adult participants, aged between 22 and 61, using a randomized controlled trial approach that incorporated SUKUBARA^®^ in the intervention. The results showed significant improvements in fat-free mass (FFM), knee extension strength (KES), and one-leg standing (OLS) with eyes closed in the intervention group as compared with the control group [17]. However, generalizing those results to sarcopenia is not appropriate as the outcomes cannot be directly applied across different age groups. Therefore, this study aimed to verify whether an intervention utilizing SUKUBARA^®^ increases community-dwelling older adults’ muscle hypertrophy and improves their physical abilities. Additionally, to examine the actual intervention effects, we investigated the relationship between changes in muscle hypertrophy and strength enhancement.

## 2. Materials and Methods

### 2.1. Participants

The healthy older individuals included in this study were recruited between February and March 2023. Posters were displayed and flyers were distributed at community centers within the city, which invited individuals to participate. The criteria for selecting participants were as follows: between the ages of 65 and 75 years; not diagnosed with any diseases that limit their ability to walk or conduct daily activities; not engaging in regular physical activity (exercising at least twice a week for 30 min or more) for a year or longer; have their own smartphone that allows them to watch YouTube videos at their discretion; know how to use a smartphone to play YouTube videos and use its features; and able to watch YouTube videos for approximately 15 min daily on their smartphone for a 12-week period (with the individual responsible for data usage costs). 

Applicants were randomly assigned to two groups: an intervention group that utilized the system (SUKUBARA^®^) and a control group that did not, using a randomization table. The intervention lasted for 12 weeks. The control group was informed to continue their daily activities as before during the study period. During recruitment, the participants were informed, verbally and in writing, that they could withdraw from this study at any time without facing any consequences. This study’s objectives and procedures were explained, and signed consent forms were obtained from all participants before this study commenced. The protocol for this study was approved by the Institutional Review Board of the University of Tsukuba Hospital (Approval Number R04-232), and this study was registered in a clinical trial database (UMIN000050290).

### 2.2. SUKUBARA^®^

One of the key features of the SUKUBARA^®^ exercise protocol is its incorporation of compound exercises that combine resistance and balance movements, offering a comprehensive approach to address frailty and sarcopenia [18]. Moreover, it has been reported that resistance exercises alone have limited effectiveness in fall prevention protocols, necessitating the addition of activities such as walking practice and functional movement training [19].

#### 2.2.1. Exercise Protocols

The exercise protocols provided by SUKUBARA^®^ comprise two types of exercises: (1) slow squats—a resistance exercise utilizing body weight—and (2) OLS—a balance exercise (Figure 1). A 30-second rest period is included between the slow squat and OLS exercises. LLRT is applied to the slow squats, wherein the participant sustains continuous flexion of the hip and knee joints, gradually lowering the hips from a standing position over a period of 10 s. This provides a centrifugal contraction load to the quadriceps muscles. Moreover, isometric contraction loads are applied to the quadriceps muscles by maintaining each joint position for 2 s after the initial 10 s. Slow squats were performed in three sets of 10 repetitions, with a rest period of 12 s between sets. Regarding the degree of knee joint flexion, the participants were instructed to maintain an angle perceived as moderately intense when the exercise intensity felt slightly challenging. The instructed range was from a minimum of 15 degrees to a maximum of 90 degrees. For OLS, participants performed the exercise by standing on one leg for 60 s, with one lower limb raised and their eyes closed. If the participant felt highly unstable, they were allowed to support themselves by grabbing onto a table or similar surface with the raised opposite upper limb. All participants in the intervention group visited the physical therapy room at our clinic for the initial session (approximately 15 min in duration), where they received instructions on how to watch the videos and perform the exercises from the same physical therapist. The recommended frequency for using SUKUBARA^®^ was a minimum of three times per week, although the participants were encouraged to perform the exercises daily whenever possible. Additionally, the intervention group was instructed not to engage in any exercises other than those provided by SUKUBARA^®^ during the study period, whereas the control group was asked not to initiate any new exercise routines.

#### 2.2.2. Monitoring System Using YouTube Studio

The videos for the exercise protocols were created and uploaded to a dedicated YouTube channel in individualized versions, each tagged with the participants’ ID numbers. The videos were then set to private using YouTube’s editing application (YouTube Studio), thus allowing limited access for each specific video. Furthermore, the participants were given a paper copy containing QR codes converted from the playback URLs. They were instructed that during their training sessions, they could use the camera function on their smartphones or tablet devices to scan the QR code assigned to them, enabling them to play their designated videos. By utilizing YouTube Studio, it was possible to monitor the total video playback time for each individual in real time, which was considered equivalent to the individual’s total training time, establishing a system to manage and monitor training duration effectively. The general SUKUBARA^®^ video is available to the public as a video file (with a total duration of 14 min and 30 s).

### 2.3. Clinical Data and Laboratory Tests

The primary endpoint of this study was a change in FFM. The secondary endpoints were changes in KES, grip strength, body fat mass volume, weight, OLS with eyes open and closed, the index of postural stability (IPS) and the modified index of postural stability (mIPS) [17,20], and the amount of physical activity during the intervention period. Of these, body composition, KES, grip strength, IPS, and mIPS were assessed once each, whereas OLS was performed twice for both the left and right sides, and the maximum value (up to 60 s) was considered. A rest period of approximately 1 min was provided between trials for OLS. Furthermore, a rest period of about 5 min was given between each assessment. The body composition of each participant was measured using Bioelectrical Impedance Analysis (BIA; InBody 720, InBody Japan, Tokyo, Japan). Although Dual Energy X-ray Absorptiometry (DEXA) is considered the gold standard in practical measurements of SMM as a component of body composition, FFM measured by BIA is also suitable [21,22,23]. Therefore, we used FFM measured by InBody as an indicator in this study. Additionally, given that SUKUBARA^®^ (squatting exercise) not only activates muscles in the lower limbs but also in the core [24], we used the whole-body FFM. The measurement time was fixed at 9:00 am (in a fasting state, i.e., no food intake since 9:00 PM the previous night) to account for the influence of environmental factors, such as meal intake and time of day. The measurement time of muscle strength or muscle/fat mass volume and balance functions before and after participating in this study was the same for each participant. KES on the dominant leg (Nm/kg) was measured using a torque machine (isokinetic strength measurements; 60°/s; Biodex System 3: Sakai Medical, Tokyo, Japan). The grip strength of the dominant hand was measured using a Smedley analog grip meter (ST100 T-1780, Toei Light Co., Tokyo, Japan). Balance capabilities were assessed based on the OLS with eyes open and closed, IPS, and mIPS. IPS and mIPS were measured using a gravicorder (GP-6000, Anima Co., Tokyo, Japan) [20]. Physical activity during the intervention period was measured using an accelerometer (Mediwalk: Terumo Co., Tokyo, Japan). Data on the average number of steps, average moderate-to-vigorous physical activity (MVPA) time (three metabolic equivalents or more), and energy expenditure through exercise were collected. These measurements were conducted in accordance with prior research [17].

### 2.4. Statistical Analysis

For all the factors evaluated, the normality of the data was assessed using the Shapiro–Wilk test. After confirming the distribution, appropriate statistical methods were chosen; variables found to have a normal distribution were presented as mean ± standard deviation, whereas variables with a non-normal distribution were presented as medians (interquartile range, 25th percentile, 75th percentile). Comparisons between groups of participant characteristics before the intervention, including age, biological sex, height, weight, body mass index, lifestyle factors (employment, alcohol consumption, history of falls), body composition, muscle strength, balance, cognitive function, and physical activity during the intervention period (step count and MVPA), were conducted using independent t-tests (for variables with normal distribution) or Mann–Whitney U tests (for variables with non-normal distribution) for continuous variables and chi-square tests for categorical variables. The comparison of interventional changes in body composition, muscle strength, and balance capability between the two groups was performed using independent t-tests or Mann–Whitney U tests (for variables with normal and non-normal distribution, respectively). For variables demonstrating a normal distribution, a two-way analysis of variance (ANOVA) including group (intervention and control groups) and time (pre- and post-intervention) was used to examine the effects of and interaction between group and time. Bonferroni correction was applied following the two-way ANOVA. For variables with a non-normal distribution, comparisons were made using the Wilcoxon signed-rank test. If differences in the change in FFM were observed between groups, we intended to examine the correlation with KES. If the factor exhibited normality, a partial correlation analysis adjusted for age and biological sex was performed; if normality was not observed, Spearman’s rank correlation coefficient was used. This study was initially designed to detect an effect size of 0.20 in the increase in FFM, with a significance level of 5% and a power of 80%. It was estimated that a minimum of 40 participants would be necessary to detect differences between groups. Statistical analyses were performed using SPSS version 24.0 (IBM Japan), and the significance level was set at 5%. 

## 3. Results

Although this study initially aimed to recruit 40 participants, only 15 individuals applied. This study had no dropouts; thus, the analysis included all 15 participants (8 in the intervention group and 7 in the control group; Figure 2). The total video playback time for the intervention group was 18.7 ± 5.6 h.

The participants’ basic attributes, lifestyle factors, body compositions, muscle strengths, balance abilities, cognitive functions before the intervention, and physical activities during the intervention period are summarized in Table 1. No significant differences were observed in any of the variables between the two groups. The intergroup comparison of changes during the intervention period and intragroup comparisons before and after the intervention are presented in Table 2 and Table 3. Table 2 presents the results of body composition and muscle strength. During the intervention period, not all the variables changed over time; however, a significant difference in the FFM (0.5 ± 0.5 vs. −0.1 ± 0.5 kg, *p* = 0.028) and KES (0.20 ± 0.22 vs. 0.02 ± 0.13 Nm/kg, *p* = 0.037) changes was observed between the groups. None of the factors showed significant interactions (group × time). Table 3 presents the results of balance capability. Since all factors were not normally distributed, interactions were not sought, and separate analyses were conducted within and between groups. During the intervention period, a significant increase was observed in OLS with eyes open (3 s, *p* = 0.018) and OLS with eyes closed (2 s, *p* = 0.034), but there were no significant differences between the groups.

As the changes in FFM and KES followed a normal distribution, partial correlation analyses adjusted for age and biological sex were conducted. The results showed a significant positive correlation between changes in FFM and KES (R = 0.557, *p* = 0.048; Figure 3).

## 4. Discussion

In this study, community-dwelling older adults were randomly assigned either to the intervention or control group, and the effects of SUKUBARA^®^ on their muscle hypertrophy and physical abilities were examined. Significant changes in body composition and physical function, particularly in FFM and KES, were observed in the intervention group compared with the control group. It is noteworthy that despite being entirely home-based and self-guided, the intervention with SUKUBARA^®^ has the potential to increase important assessment parameters, such as FFM and KES, among community-dwelling older adults. YouTube, which is integrated into the SUKUBARA^®^ system, comes pre-installed as a default Google application on commercially available smartphones, and is, thus, widely available, convenient, and free. Therefore, SUKUBARA^®^ is highly suitable for extensive and large-scale dissemination as a training system for community-dwelling older adults. 

Our findings further indicate a moderate positive correlation between the changes in FFM and KES by interventions. In previous reports, interventions involving resistance training have shown a positive correlation between an increase in skeletal muscle volume or anatomical muscle cross-sectional area and muscle strength [25,26]. Interventions with SUKUBARA^®^ may provide the effects of both muscle hypertrophy and strength enhancement without dissociation. Moreover, the training period required for muscle hypertrophy is generally considered to be 12 weeks or more. However, in this study, an improvement in FFM was observed in less than 12 weeks. According to recent systematic reviews, when aiming for effective muscle hypertrophy using resistance training, designing training based on the total load (frequency × duration) is recommended [27]. In the intervention group, the average total playback time for participants was 18.7 h. Considering the completion time for each program as 14.5 min, this translates to a frequency of 6.2 sessions per week (over 12 weeks). As a result, the improvement in FFM within 12 weeks may be because of the considerably high exercise frequency during the intervention period.

The results of this study regarding the improvement in participants’ muscle hypertrophy and strength can be attributed to two main factors. The first is the potential impact of the imagery effect from watching the videos in the SUKUBARA^®^ exercise protocol. In a previous study involving young participants, watching videos of high-intensity resistance exercises led to increased muscle strength in the specific muscle groups even without performing the actual exercises [28]. This phenomenon has been attributed to the effects of both motor imagery and action observation (observational learning). Moreover, in previous studies targeting older adults living independently in a community, providing a comprehensive system, including tablet-based exercise videos, was reported to be more effective in improving walking and physical abilities compared with providing pamphlets as guides [29]. In this study, participants synchronized their squat movements with the exercise protocol videos they watched, while matching the audio instructions; this synchronized approach might have amplified the training effects. The second reason lies in the adherence of the participants to the recommended frequency of the SUKUBARA^®^ sessions. Previous studies, such as Braith et al., reported a significant increase in muscle strength in a group performing knee extension exercises thrice a week compared with a group exercising twice a week [30]. This suggests that a higher training frequency may lead to more significant effects. All eight participants in the intervention group adhered to a frequency of three or more sessions per week. The adherence of the participants to the recommended frequency of SUKUBARA^®^ training could be attributed to their awareness that their daily video viewing time was being monitored in real time by the research administrators. It has been suggested that individuals may modify their behavior when they are aware of being monitored [31,32]. In other words, although not actually under direct surveillance, participants might have experienced a psychological state of feeling “remotely monitored”, leading them to diligently engage in daily training sessions.

### Limitation

There are six main limitations to this study. First, an insufficient number of applicants were recruited owing to restrictions on outdoor activities and the voluntary restraint of group activities amidst the recent COVID-19 pandemic. Additionally, a reduction in planned promotional activities occurred due to the unexpected discontinuation of a local newspaper in which the recruitment advertisements appeared. Consequently, the final number of participants was fewer than half of the initially planned sample size. Conducting another intervention study with the originally intended sample size and interpreting the results using statistical significance tests will provide a more conclusive validation. Second, there were no significant differences in the group × time interaction between the intervention and control groups in the statistical analysis. These results may be attributed to the small sample size; thus, it will be necessary to re-evaluate the results with the originally intended sample size in the future. Third, FFM was measured by the BIA method as a surrogate indicator for SMM. However, the gold standard for SMM in clinical settings is an index measured by the DEXA method. Therefore, to formally assert the muscle hypertrophy effects of this intervention, it is necessary to re-evaluate the results using the DEXA method. Fourth, no intervention effect was observed in the comparison of balance-related outcomes between the groups. A systematic review investigating the impact of supervision during resistance and balance exercises in older individuals reported that both static and dynamic balance measures show greater improvement under supervised interventions than completely unsupervised interventions [18]. Therefore, further investigation is needed to determine whether interventions such as SUKUBARA^®^, conducted without direct supervision, have an impact on the balance abilities of older individuals. Fifth, the impact of SUKUBARA^®^ implementation on the activities of daily living, care needs, or healthcare economy of community-dwelling older individuals remains unknown. At this point, it is not possible to confirm this system as an effective treatment for frailty or sarcopenia. The widespread implementation of SUKUBARA^®^ among a large population of community-dwelling older adults and an evaluation of its effects from a long-term perspective would allow for a proper assessment of its true effectiveness. Finally, despite being a preliminary study aimed at preventing sarcopenia, this research did not measure participants’ physical abilities such as walking speed or the short physical performance battery score [5]. Consequently, it was not possible to identify the status of sarcopenia. In future studies, physical performance indicators should be measured according to the latest AWGS 2019 criteria [5]. This will help in clearly identifying any improvement in the attributes of sarcopenia among the participants.

## 5. Conclusions

The home-based training system consisting of a simple compound exercise protocol and a remote monitoring system, SUKUBARA^®^, demonstrated the potential to improve muscle hypertrophy and strength in community-dwelling older adults. However, this study needs to replicate its findings in appropriately powered research in the future.

## Figures and Tables

**Figure 1 geriatrics-09-00020-f001:**
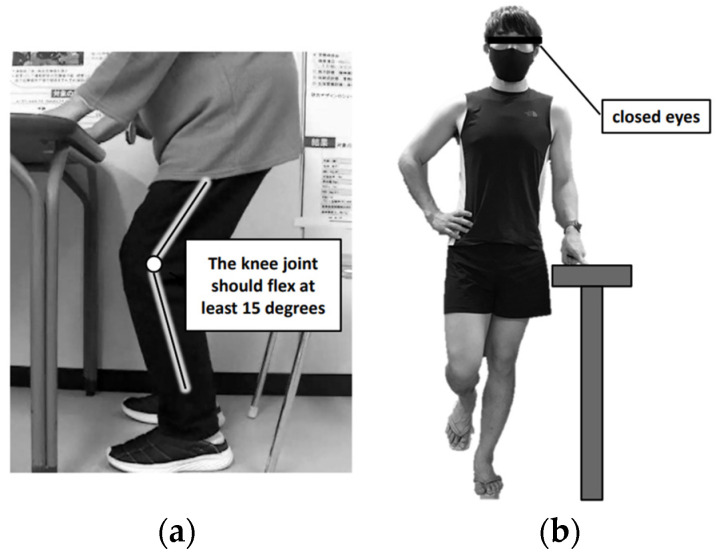
Exercise protocols provided by SUKUBARA^®^. (**a**) Resistance exercises (slow squats). (**b**) Balance exercises (one-legged stance with eyes closed).

**Figure 2 geriatrics-09-00020-f002:**
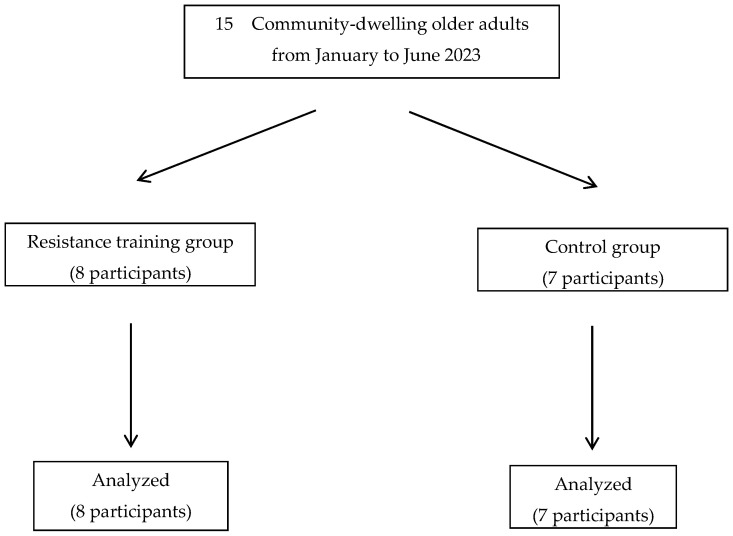
Flowchart of the selection of study participants.

**Figure 3 geriatrics-09-00020-f003:**
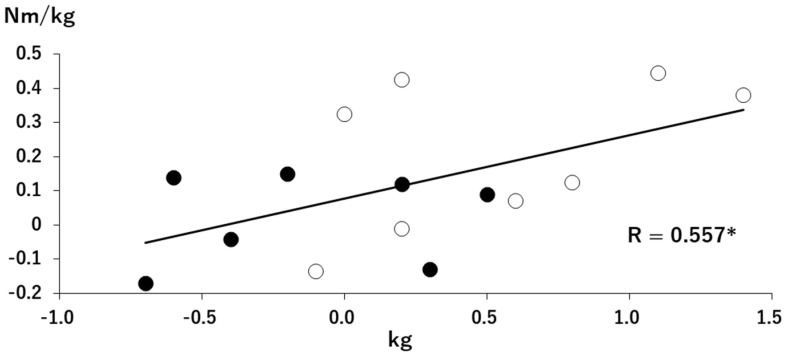
Relationship between changes in fat-free mass and knee extension strength. 〇: Resistance training group. ●: Control group. * *p* < 0.05.

**Table 1 geriatrics-09-00020-t001:** Characteristics of participants.

	RT *n* = 8	CO *n* = 7	*p*
Age (years)	67 (65, 70)	72 (66, 73)	0.407
Female, n (%)	2 (25)	3 (43)	0.480
Height (m)	1.65 (1.62, 1.71)	1.67 (1.58, 1.74)	0.953
Body weight (kg)	61.6 ± 9.6	66.5 ± 7.0	0.290
Body mass index (kg/m^2^)	22.3 ± 2.4	24.3 ± 1.3	0.079
Currently working, n (%)	2 (33)	1 (20)	0.613
Currently driving, n (%)	7 (88)	7 (100)	0.694
Fall history, n (%)	0 (0)	0 (0)	1.000
Body composition			
Fat-free mass (kg)	25.8 ± 5.0	27.1 ± 5.7	0.652
Body fat mass (kg)	14.7 ± 3.3	17.3 ± 4.6	0.225
Muscle strength			
Grip strength (kgf)	37 (31, 47)	37 (29, 46)	0.602
Knee extension strength (Nm/kg)	1.54 ± 0.45	1.62 ± 0.26	0.685
Balance capability			
Index of postural stability	1.86 (1.76, 2.01)	1.62 (1.57, 1.67)	0.073
Modified index of postural stability	0.46 (0.22, 0.60)	0.43 (0.32, 0.59)	1.000
One-leg standing time with eyes open (s)	95 (52, 120)	99 (90, 116)	0.720
One-leg standing time with eyes closed (s)	4 (3, 6)	4 (3, 6)	0.694
Cognitive function			
MMSE (points)	30 (30, 30)	30 (29, 30)	0.613
Physical activity			
Steps (steps/day) *	7279 ± 1878	8640 ± 3952	0.346
MVPA time (min/day) *	14.0 (11.6, 21.0)	23.7 (15.4, 34.7)	0.247

Data are in the form of mean ± SD or median (25th, 75th). RT, resistance training group; CO, control group; MMSE, mini-mental state examination; MVPA, moderate-to-vigorous physical activity. *: Data from during the intervention period.

**Table 2 geriatrics-09-00020-t002:** Comparison of participants’ body composition and muscle strength, within and between groups.

	Group	Baseline	12 Weeks	p for Time	Change between Baseline and 12 Weeks	p for Group	p for Group × Time
Body composition							
Body weight (kg)	RT	61.6 ± 9.6	61.6 ± 9.2	0.995	−0.02 ± 1.4	0.251	0.894
	CO	66.5 ± 7.0	65.6 ± 7.3	0.851	−0.9 ± 1.3		
Fat-free mass (kg)	RT	25.8 ± 5.0	26.3 ± 4.9	0.843	0.5 ± 0.5	0.028	0.866
	CO	27.1 ± 5.7	26.9 ± 5.5	0.964	−0.1 ± 0.5		
Body fat mass (kg)	RT	14.7 ± 3.3	13.9 ± 3.0	0.663	−0.9 ± 0.6	0.254	0.928
	CO	17.3 ± 4.6	16.7 ± 4.7	0.777	−0.6 ± 0.9		
Muscle strength							
Grip strength (kgf)	RT	37.3 ± 9.5	37.1 ± 9.5	0.977	−0.1 ± 2.5	0.398	0.651
	CO	34.6 ± 7.5	37.4 ± 7.5	0.561	1.0 ± 2.3		
Knee extension strength (Nm/kg)	RT	1.54 ± 0.54	1.74 ± 0.33	0.226	0.20 ± 0.22	0.041	0.459
	CO	1.62 ± 0.26	1.65 ± 0.17	0.897	0.02 ± 0.13		

RT, resistance training group; CO, control group; data are in the form of mean ± SD or median (25th, 75th).

**Table 3 geriatrics-09-00020-t003:** Comparison of participants’ balance capability, within and between groups.

	Group	Baseline	12 Weeks	p for Time	Change between Baseline and 12 Weeks	p for Group
Balance capability						
Index of postural stability	RT	1.86 (1.76, 2.01)	1.91 (1.76, 2.06)	0.799	0.06 (−0.11, 0.14)	0.897
	CO	1.62 (1.57, 1.67)	1.55 (1.47, 1.78)	0.917	0.01 (−0.13, 0.16)	
Modified index of postural stability	RT	0.46 (0.22, 0.60)	0.39 (0.14, 0.61)	0.779	0.04 (−0.05, 0.09)	0.365
	CO	0.43 (0.32, 0.59)	0.31 (0.18, 0.40)	0.116	−0.18 (−0.22, −0.01)	
One-leg standing time with eyes open (s)	RT	95 (52, 120)	116 (53, 120)	0.720	0 (−3, 2)	0.120
	CO	99 (90, 116)	74 (30, 120)	0.080	−25 (−50, 0)	
One-leg standing time with eyes closed (s)	RT	4 (3, 5)	7 (6, 11)	0.018	3 (1, 7)	0.726
	CO	4 (3, 6)	6 (4, 8)	0.034	2 (1, 4)	

RT, resistance training group; CO, control group; data are in the form of mean ± SD or median (25th, 75th).

## Data Availability

The data used to support the findings of this study are available from the corresponding author upon request. The data are not publicly available because they contain information that can compromise the privacy of the research participants.

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
