# Peer review of "Effectiveness of a Remote Monitoring-Based Home Training System for Preventing Frailty in Older Adults in Japan: A Preliminary Randomized Controlled Trial"

_geriatrics, 2024, doi:10.3390/geriatrics9010020_

Round 1
Reviewer 1 Report
Comments and Suggestions for Authors
The authors chose to research a current problem. It is an exciting question to research the possibilities of developing muscle strength within physical activity in old age.
Contrary to the original intention, only a relatively small number of subjects participated in the sukubara program, is a serious limitation.
The design of the experiment is correct, the methodologies used are appropriate, although bioimpedance tissue determination (SMM) is not a sufficiently accurate method in this case.
The following problems were formulated in connection with the study:
1. The 12-week intervention in old age, significantly prolongs the learning phase, hypertrophy appears only in the second phase.
2. Not sure how the workout program and hand grip force are related?
3. Inbody 720 is suitable for estimating muscle mass of the lower limb (ROI).
4. There was no comparison between muscle mass growth and measured muscle strength.
5. The low number of sampla prevents evidence that training time and adaptation can be significantly related at this age (this is indicated from the partial correlation results).
5. One of the great value of the work is that, it ensures the self-control of the intervention with a witty procedure, the method can be recommended more widely.
Author Response
|
Comments 1: The 12-week intervention in old age, significantly prolongs the learning phase, hypertrophy appears only in the second phase. |
|
Response 1: Thank you for highlighting this. As you rightly point out, the duration of resistance training typically required for muscle hypertrophy is considered to be 12 weeks or more. However, because of the high frequency of exercise implementation during the intervention period in this study, it was hypothesized that improvements in FFM could occur within the 12-week timeframe. I have accordingly added the relevant information to the manuscript for greater clarity on this point (p.9, line 311–320). |
|
Comments 2: Not sure how the workout program and hand grip force are related? |
|
Response 2: Thank you for your comment. The exercise program implemented in this study focuses on training the lower limbs and core, and is believed to have minimal impact on the upper limbs. However, there are reports suggesting that observational learning induces mental visualization of the exercise, leading to improved strength (Ref. 18). Therefore, as a precautionary measure, grip strength measurements were conducted, and the changes after the intervention have been included in the results. (Tables 1, 2). |
|
Comments 3: Inbody 720 is suitable for estimating muscle mass of the lower limb (ROI). |
|
Response 3: Thank you for your comment. Although, Dual Energy X-ray Absorptiometry (DEX) is considered the gold standard in practical measurements of SMM as a component of body composition, attention is also given to FFM measured by Bioelectrical Impedance Analysis (BIA). The FFM measured by InBody shows good compatibility with SMM measured by DEX, and allows for the easy acquisition of information on body composition in a short time (less than 3 minutes) with the participant standing. Therefore, in this study, we used FFM measured by InBody as an indicator. Additionally, given that the SUKUBARA® (squatting exercise) not only activates muscles in the lower limbs but also in the core, we adopted the whole-body FFM (p.4, line 184-192). |
|
Comments 4: There was no comparison between muscle mass growth and measured muscle strength. |
|
Response 4: Thank you for highlighting this. In response to your suggestion, we re-evaluated the correlation between both factors, and were able to identify a correlation between the two. We have accordingly revised the relevant section (Figure 3, p.8, line 272–275). |
|
Comments 5: The low number of sampla prevents evidence that training time and adaptation can be significantly related at this age (this is indicated from the partial correlation results). |
|
Response 5: Thank you for rightly highlighting this. In accordance with your suggestion, I have removed this result. |
|
Comments 6: One of the great value of the work is that, it ensures the self-control of the intervention with a witty procedure, the method can be recommended more widely. |

Reviewer 2 Report
Comments and Suggestions for Authors
This is an interesting article based on RCT in Frailty among an older adults in Japan. The design of the study is appropriate. It is mentioned that a non-normal distribution exists for outcome variable therefore non-parametric analysis was used. However, you also used partial correlated adjusted for other variables. What type of correlation is this? Partial or rank but nothing was mentioned.
Author Response
|
Comments 1: The design of the study is appropriate. It is mentioned that a non-normal distribution exists for outcome variable therefore non-parametric analysis was used. However, you also used partial correlated adjusted for other variables. What type of correlation is this? Partial or rank but nothing was mentioned. |
|
Response 1: Thank you for highlighting this. Based on your comment, the initially planned partial correlation analysis using SMI, KES, and OFS was abandoned, and a new analysis of the partial correlation between FFM and KES was conducted. Since both factors are normally distributed, we believe there is no issue in performing a partial correlation analysis. Accordingly, we have revised the methods and results sections (p.9, line 303–311). |

Reviewer 3 Report
Comments and Suggestions for Authors
Summary: The aim of this manuscript by Suzuki et al. was to investigate whether an intervention utilizing sukubara® among community-dwelling older adults leads to an increase in skeletal muscle mass and improvement in physical abilities. Additionally, this study aimed to assess whether the practical management for the system, and the authors examined the relationship between total video playback time on YouTube, activity records during the intervention period, and pre- to post values for several outcomes. The main contribution of this manuscript is that it adds to the literature by investigating the effectiveness of a remote monitoring-based home training system on fat-free mass, strength, and balance in community-dwelling older adults. The results showed that participants in the intervention group increased fat-free mass and knee extension strength with positive correlations between total video playback time and fat-free mass and knee extension strength. Strengths of this study are that it compares the effectiveness of a remote monitoring-based home training system in community dwelling older adults using a randomized controlled trial design. In addition, a notable strength of this manuscript is the appropriate interpretations and conclusions made by the authors based on the data.
General comments: Overall, this study was generally well conducted and written, with appropriate interpretations and conclusions based on the results of the data. This manuscript adds to the literature by investigating the effectiveness of a remote monitoring-based home training system on fat-free mass, strength, and balance in community-dwelling older adults. This information and the use of such a remote monitoring-based home training system which can be implemented with relative ease and is scalable can help researchers, practitioners, public health officials, and those working in healthcare with older adults to utilize or develop the same or similar remote monitoring-based home training systems to improve older adults fat-free mass and measures of strength. A concern of the manuscript is the sample size of participants of 15 compared to a priori sample size calculation of 40 participants needed to detect to an effect size of 0.20 increase in skeletal muscle mass with a significance level of 5% and a power of 80%. However, the authors do note and address this issue in the limitations section and state the need for conducting this study in a larger sample size.
Specific comments:
General comment throughout regarding wording:
· The use of the wording skeletal muscle mass and acronym SMM should be changed to fat-free mass and the acronym FFM throughout because BIA is a two-compartment model that estimates fat mass and fat-free mass, but not specifically skeletal muscle mass, and fat-free mass includes body water, organs, bone, and muscle.
Line 141: There should be a space between the last word in the sentence and references cited.
Lines 206-229: Section 2.3 Clinical Data and Laboratory Tests: Was grip strength measured both pre- and post-intervention or just at baseline? It appears as if it was only measured at baseline. Please add a statement if it was only measured at baseline or if it was measured pre- and post-intervention and was a secondary outcome.
Also, how many times were KES, grip strength, OFS, and IPS assessed. If more than once, how much rest did participants have between each attempt and between assessments.
Line 125: SMM is in red font and should be in black font.
Line 217: I assume the word “valance” should be “balance”.
Line 256: SMM is in red font and should be in black font.
Line 350: Citation number “22” is in red font and should be black font.
Comments on the Quality of English Language
Minor grammatical errors to fix.
Author Response
|
Comments 1: The use of the wording skeletal muscle mass and acronym SMM should be changed to fat-free mass and the acronym FFM throughout because BIA is a two-compartment model that estimates fat mass and fat-free mass, but not specifically skeletal muscle mass, and fat-free mass includes body water, organs, bone, and muscle. |
|
Response 1: Thank you for highlighting this. We have accordingly made corrections throughout the manuscript, changing Skeletal Muscle Mass (SMM) to Fat-Free Mass (FFM) measured using the BIA method. |
|
Comments 2: Line 141: There should be a space between the last word in the sentence and references cited.  |
|
Response 2: Thank you for your careful review, we have corrected this. |
|
Comments 3: Lines 206-229: Section 2.3 Clinical Data and Laboratory Tests: Was grip strength measured both pre- and post-intervention or just at baseline? It appears as if it was only measured at baseline. Please add a statement if it was only measured at baseline or if it was measured pre- and post-intervention and was a secondary outcome. |
|
Response 3: Thank you for highlighting this point. Grip strength was measured both before and after the training sessions, but it was not clearly described in the previous version of the manuscript. It has been added to the table and methods (secondary outcome measures) for clarity (Table 2, p.4 line 176).
|
|
Comments 4: Also, how many times were KES, grip strength, OFS, and IPS assessed. If more than once, how much rest did participants have between each attempt and between assessments.  |
|
Response 4: Thank you for bringing this to our notice. We conducted KES, grip strength, IPS, and mIPS measurements once each, and OFS measurements twice for both left and right sides. For OFS, approximately 1 minute of rest was provided between trials. Additionally, there was a rest period of about 5 minutes between each assessment. We have included this information in the relevant section (p.4, line 178–182).
|
|
Comments 5: Line 125: SMM is in red font and should be in black font. |
|
Response 5: Thank you for your careful review. We have made the correction.
|
|
Comments 6: Line 217: I assume the word “valance” should be “balance”. |
|
Response 6: Thank you for highlighting this, we have corrected the word.
|
|
Comments 7: Line 256: SMM is in red font and should be in black font. |
|
Response 7: Thank you, we have made this correction.
|
|
Comments 8: Line 350: Citation number “22” is in red font and should be black font. |
|
Response 8: Thank you for your detailed comments, we have made this correction.
|
